# The Relationship between Exercise Re-Participation Intention Based on the Sports-Socialization Process: YouTube Sports Content Intervention

**DOI:** 10.3390/bs13020187

**Published:** 2023-02-18

**Authors:** Youngtaek Oh

**Affiliations:** Department of Kinesiology & Sport Management, Texas A&M University, College Station, TX 77843, USA; oyt0712@tamu.edu; Tel.: +1-979-307-8471

**Keywords:** exercise performance satisfaction, exercise interruption intention, re-participation intention, social support, YouTube sports content

## Abstract

Few studies have used a quantitative research methodology to examine the socialization process model, and such studies were conducted to verify a new model by intervening in the variables of YouTube sports content. To understand this process, we tested the moderated mediating effect by intervening in YouTube sports content based on the sports socialization process model. We recruited 274 participants from the Jeju Residents’ Jeju Sports Festival, Korea. The PROCESS Macro program was performed to test the research hypotheses. The findings indicate that social support had a significant effect on re-participation intention. Social support had a significant mediation effect on exercise interruption intention, re-participation intention, and exercise performance satisfaction. Furthermore, through the relationship between social support and exercise interruption intention, YouTube sports content showed a significant interaction of re-participation intention in exercise. These results extend sports socialization theory by discovering a new model that explains the relationship between the sports socialization process and YouTube sports content. In addition, it will provide a basis for delivering educational information to the public so that they can recognize the importance of physical activity and exercise skills.

## 1. Introduction

The sports socialization process provides a general framework for participating in and performing physical activities with various socio-psychological influences. It also introduces information on how socialization is approached theoretically and provides a course on the nature of socialization and how to perform [1]. When human beings experience sports socialization, they experience the processes of socialization into sport, socialization via sport, desocialization from sport, and resocialization into sport.

People have different motives for experiencing the process of socialization in sports. In particular, after experiencing the process of desocialization from sport, the motivation for resocialization into sport will appear differently depending on a person’s psychological needs [2]. Based on the sports socialization process model, this study will analyze the moderating effect of YouTube sports content on the mediating effects of exercise performance satisfaction and exercise interruption intention in the relationship between the sport psychological variables of social support and re-participation intention.

Socialization is the process by which individuals internalize the skills, knowledge, and roles that are appropriate given their place in society [3]. People who promote socialization in an individual’s environment are motivated by their family, friends, school, and community [4,5]. Socialization also includes the symbolic and cultural meanings that individuals represent in sporting societies [6]. An understanding of the process of socializing people into sports requires an understanding of the interaction of socialization [7]. Thus, many researchers are working to develop a theoretical understanding of the various aspects of sport socialization [8]. However, most previous studies explaining the process of sports socialization used qualitative research methods [8,9,10,11]. There is a deficiency in the field of social sciences, as no studies have attempted to confirm the relationships between environmental variables using quantitative research methods. This study aims to improve the understanding of YouTube content by providing a theoretical explanation of people’s perceptions and exercise participation throughout the sports socialization process. The present study focused on explaining the mechanism of the sports socialization process model by applying a quantitative research methodology to the participants during physical activity.

In this study, YouTube sports content intervened in the sports socialization process model and confirmed the interaction effect. YouTube is a resource for watching videos containing very diverse content, and it attracts a vast range of visitors worldwide [12]. For example, politicians, media outlets, educational institutions, corporations, music and film artists, and people from all walks of life consume content on YouTube. Even for people who engage in physical activity, YouTube content has become an important medium for understanding training methods. Although some studies report that watching YouTube has a negative effect [13,14], a recent study reports that YouTube activity plays a positive role in fitness participants [15,16,17], sports development [18], and participation in physical activity. The research model will confirm the two-way (positive/negative) nature of YouTube participation.

This study will contribute to the theory of the sports socialization process by confirming the paths of satisfaction with exercise performance satisfaction and exercise interruption intention in the relationship between social support and re-participation intention. By considering the intervention of YouTube sports content, it will also explain the importance of YouTube content in human social psychology and the role of social media in re-participation intention in sports in people’s social lives. In particular, satisfaction with exercise performance after exercise participation and exercise interruption intention are expected to promote exercise re-participation intention. The purpose of this study is to confirm the multi-mediating effects of satisfaction with exercise performance satisfaction and exercise interruption intention in the relationship between social support and re-participation intention. It also aims to confirm the interaction of sports content on YouTube.

### 1.1. Literature Review

#### 1.1.1. Sport Socialization Model

This study was conducted based on the sports socialization process model. The process of sports socialization involves sequentially experiencing socialization into sport, socialization via sport, desocialization from sport, and resocialization into sport. Socialization into sport is primarily related to an individual’s participation in sports, which is influenced by personal characteristics, key players, and external factors [19]. In particular, socialization into sport generally involves one’s parents, teachers, coaches, peers, siblings [20], role aspirants, and the social context in which learning takes place [21]. For example, it can appear differently depending on the individual’s internal/external satisfaction, social cohesion, and sports identity, as well as the time, money, and energy they invest in sports. Socialization into sports appears differently depending on one’s age, gender, and birth order as innate factors and their education level, social class, religion, and residential area as acquired factors. However, the literature shows that over-emphasizing athletic participation can lower an individual’s academic achievement, increase their expectations of having a professional sports career, reinforce their athletic identity, and reduce the likelihood they will have an athletic career [22,23,24]. This process develops as socialization is presented to children by adults. 

The next step is socialization via sport, which is the psychological experience that one has after directly participating in sports. Socialization via sport is related to the outcome of a match and sportspersonship [25]. Furthermore, it provides an opportunity for participants to understand the value of sports through morality [26] and aggression [27] while acquiring knowledge, attitudes, norms, and behaviors necessary for a healthy social life. For example, when individuals exercise, they are conscientious about body management [27], loyalty [28], and skill improvement [29].

While participating in sports, people experience desocialization from sport or resocialization into sport owing to various environmental factors. For example, socialization from sport may occur due to loss of organizational stress, exercise identity, social relationships, daily structure, support, and the sense of purpose felt while participating in exercise [30] or retirement [31]. Meanwhile, desocialization from sport is experienced after involuntary retirement, which may be caused by an injury, physical changes, career dissatisfaction [32], or interest in activities other than sports [32]. In other words, desocialization from sport is a situation in which exercise is stopped. However, people who undergo desocialization from sport do not refrain from participating in sports altogether for the rest of their lives, as the resocialization into sport process provides an opportunity to participate again. The process of re-participation includes the phenomenon of participating in sports again with new motivation after one has stopped participating in sports [19]. It often entails directly participating in sports or engaging in sports-related activities (e.g., coaching, commentating, or administrating). 

This study was conducted based on the sports socialization process model (Socialization into sport → Socialization via sport → Desocialization from sport → Resocialization into sport). The multi-mediating effect of exercise performance satisfaction and exercise interruption intention in the process of social support and re-participation intention was confirmed using sports socio-psychological terms. Furthermore, interactions were identified through the intervention of YouTube sports content. The detailed research concept and review of previous studies are described in the following sections.

#### 1.1.2. Relationship between Social Support and Exercise Re-Participation Intention

This study seeks to determine how and to what extent sports participation affects the formation of a model that meets specific human needs during the sports socialization of physical activity participants. In the first step for analysis, we will confirm the relationship between social support and re-participation intention in exercise (H1). Social support is a multidimensional construct that includes various forms of help from significant others or the external environment, the structure of an individual’s social network, and the resources available from one’s interpersonal relationships [33]. Social support, the first step in the model, motivates people to exercise. Thus, exercise begins with social support. People experience psychological well-being [34] and positive psychology approach [35] after participating in exercise. Hypothesis 1 will be tested to validate the model.

This study explains whether physical activity participants can continue to participate in physical activity by utilizing social support resources. Participants who experience exercise re-participation intention feel various psychological emotions after participating in a specific activity [36]. While experiencing the activity, they perceive satisfaction [37] and happiness [38] and feel various positive emotions about the activity, which increases their intention to continue participating and re-participating. Previous research results on participants in fitness [39], yachting [37], and trail running [40] indicate that the intention to re-participate in exercise increases over time. In other words, physical activity participants strengthen their re-participation intention through social support. Based on previous studies, Hypothesis 1 was developed.

**Hypothesis** **1** **(H1).** 
*Social support significantly affects re-participation intention.*


#### 1.1.3. Relationship between Social Support, Exercise Performance Satisfaction, Exercise Interruption Intention, and Re-Participation Intention

Social support among elite athletes in a sports environment is actively studied [41,42,43]. Specifically, this study emphasizes the importance of observing Korean adults who participate in exercise daily. This is done by examining the relationship between social support, exercise performance satisfaction, exercise interruption intention, and re-participation intention. According to previous research, social support positively affects exercise performance satisfaction [41,44,45,46], which is one’s ability to maximize their skills and abilities by investing time and effort in a sport [47]. People who perceive that they receive social support have shown increased exercise performance satisfaction because they actively participate in exercise participation, both psychologically and technically [45]. 

Exercise interruption intention refers to a state in which a person does not want to participate in exercise anymore and can occur when personal, psychological, and socio-environmental factors such as discomfort, stress, and exhaustion are experienced while participating in exercise [48]. In Moon’s [49] study, social support reduced exercise cessation. Other researchers have found that exercise performance satisfaction is positively related to exercise continuation [50], self-efficacy [51], psychological well-being [52], and intention to re-participate [53]. Meanwhile, many studies have verified chain relationships between social support and exercise performance satisfaction [41] and between exercise performance satisfaction and re-participation intention [54]. However, no study has confirmed the relationships among all variables based on the sports socialization process model. Based on previous studies, the following hypotheses were developed. 

**Hypothesis** **2** **(H2).** 
*Social support has a significant indirect effect on re-participation intention in exercise via exercise interruption intention.*


**Hypothesis** **3** **(H3).** 
*Social support has a significant indirect effect on re-participation intention in exercise via exercise performance satisfaction.*


**Hypothesis** **4** **(H4).** 
*Social support has a significant effect on re-participation intention by mediating the relationship between exercise performance satisfaction and exercise interruption intention.*


#### 1.1.4. YouTube Sports Content

YouTube was founded as youtube.com (YouTube, n.d.), a private video-sharing service [55]. This platform was advantageous in that it allowed users to provide their own content to many consumers via the Internet. In particular, one’s thoughts on the content image can be displayed. YouTube also enables users to share multiple comments and videos and provides viewers with buttons they can click to indicate whether they like or dislike a video [56]. As users can provide and create content, the YouTube model of developing content that offers various types of information has become tremendously popular [57].

In particular interest to the present study, YouTube sports content is also constantly evolving. Shen et al. [17] described fitness YouTube videos as demonstrating clarity and expertise to help arouse users’ interest in achieving learning outcomes. Watching sports videos on YouTube has the advantage of improving the user’s motor skills more effectively than conventional learning methods. Studies on YouTube videos have shown that videos about the technical abilities used in soccer [56], basketball [58], and volleyball [59] are receiving much attention from participants. Additionally, YouTube has systems for team analysis [60,61], promotion [62], and injury treatment [63]. YouTube is continually trying to discover its effectiveness in education [64,65]. As such, YouTube provides a variety of positive information to athletes. Despite this, no study in the field of sports sociopsychology has verified the relevance of YouTube sports content using the sports socialization process model. Therefore, Hypothesis 5 was proposed.

**Hypothesis** **5** **(H5).** 
*Social support, exercise performance satisfaction, and exercise interruption intention significantly affect re-participation intention through the interaction of YouTube sport content.*


The research model is as follows, Figure 1.

## 2. Method

### 2.1. Participants

The participants who completed the survey in this study were 274 adults who participated in the Jeju Sports Festival in Korea (1 to 7 December 2022). The Jeju Sports Festival was held on 28 to 30 October 2022; however, this study was conducted after the sports festival. The sample was comprised of 204 males (74.5%) and 70 females (25.5%). Of the participants, 140 (51.1%) were in their 20s, 96 (35.0%) were in their 30s, and 38 (13.9%) were in their 40s.

### 2.2. Measures

#### Questionnaire Scales

The questionnaire was comprised of scales of reliability and validity which have been deemed adequate in previous studies. Social support was measured with the 12-item scale [66], which was based on a previous scale [67]. We used 5 items from Park [47] to measure exercise performance satisfaction, 10 items pertaining to exercise interruption intention from Park et al. [68], 5 items pertaining to exercise re-participation intention were adopted from Lee [69], 17 items pertaining to YouTube sport content were adopted from Rubin [70]. All items were measured on a 5-point Likert-type scale (1 = strongly disagree; 5 = strongly agree). Selected items were modified according to the participant respective contexts based on experts’ suggestions pertaining to content relevance and item clarity.

### 2.3. Statistical Analysis

The collected data were analyzed using SPSS 24.0, SPSS PROCESS Macro, and Amos 24.0 statistical programs with an alpha level of 0.05. First, a frequency analysis was conducted. Second, Cronbach’s alpha values were calculated to check the reliability of each measurement tool, and confirmatory factor analysis was performed to ascertain the validity of the constructs. Third, Pearson’s product–moment correlation was calculated on major variables. Fourth, SPSS PROCESS Macro (Model no. 6, 89; Hayes [71]) was used to analyze the moderated mediation effect. Finally, the conditional indirect effect was confirmed.

## 3. Results

### 3.1. Result of Statistical and Correlation Analyses

The descriptive statistics of study variables (including mean, standard deviation, skewness and kurtosis, composite reliability, and average variance extracted) are listed in Table 1. The data showed a normal distribution, as the values of skewness and kurtosis fell within the recommended ranges—from −2 to +2 and from −7 to +7, respectively [72]. Correlations were calculated to examine overall relationships between variables, and all variables were found to be correlated below 0.70 (see Table 1), confirming an absence of multicollinearity [73].

### 3.2. Direct and Indirect Effects of Social Support, Exercise Performance Satisfaction, Exercise Interruption Intention, and Re-Participation Intention

Social support had a significant positive effect on re-participation intention; thus, Hypothesis 1 was accepted. Furthermore, social support had a significant positive effect on exercise performance satisfaction, while exercise performance satisfaction had a significant negative effect on exercise interruption intention. Moreover, exercise performance satisfaction had a significant positive effect on re-participation intention, and exercise interruption intention had a significant negative effect on re-participation intention (see Table 2).

Since several indirect effects were confirmed, the total effect was significant. Specifically, the indirect effects of social support, exercise interruption intention, and re-participation intention were significant; thus, Hypothesis 2 was accepted. Furthermore, the indirect effects of social support, exercise performance satisfaction, and re-participation intention were also significant, meaning that Hypothesis 3 was accepted. However, no indirect effect was found for social support, exercise performance satisfaction, exercise interruption intention, and re-participation intention; thus, Hypothesis 4 was rejected.

### 3.3. Interaction Validation of YouTube Sports Content

The interactions between social support and exercise interruption intention showed significant interactions with YouTube sports content. First, when the interaction between social support and YouTube content was low, medium or high, it had a significant effect on re-participation intention. Second, the interaction between exercise interruption intention and YouTube content was significant at high and medium levels. Hypothesis 5 was partially accepted (see Table 3, Figure 2).

## 4. Discussion

Human beings can participate in physical activities through specific socialization processes, namely, socialization into sport, socialization via sport, resocialization into sport, and desocialization from sport. However, people do not necessarily stop engaging in a sport for the rest of their lives after they stop playing. For example, the external environment (e.g., YouTube) allows people to participate in physical activities or engage in sports-related work via the process of sports socialization.

This study described the design of a research model based on the sports socialization process intended to confirm the visual effect of YouTube sports content and the role of environmental variables that enable people to continuously engage in physical activity. YouTube sports content is an important variable in physical activity according to studies by Sokolova and Perez [16] and McDonough et al. [74]. The purpose of the present study was to confirm the path of exercise performance satisfaction and exercise interruption intention in the relationship between social support and exercise re-participation intention. This study also intended to confirm the interaction of YouTube sports content. An in-depth discussion of the results follows. Social support significantly affected re-participation intention (H1). This outcome corroborates a study on 459 Chinese university students showing that social support had a significant effect on the intention to continue exercising [75]. Furthermore, Kim [76] studied 428 taekwondo athletes and revealed that coaches’ support had a significant effect on the players’ continuation of exercise. In other research, Lim and Wang [77] examined 701 middle school students and found that physical activity had a significant effect on intention, thus indirectly supporting the results of this study.

As a result, it can be concluded that the desire to continuously participate in exercise increases when one performs exercise with people with whom they are close [78]. For example, a close companion can provide encouragement to actively participate in exercise through positive feedback during exercise [79]. In other words, the desire to re-participate in exercise increases when information on physical activity, companions, and facilities are available to support one’s exercise participation. The current finding that social support had a significant indirect effect on re-participation intention in exercise via exercise interruption intention (H2) is supported by previous research. For instance, Park et al. [68] studied 280 kumdo players and found that parental social support had a significant effect on exercise interruption intention. The same study emphasized the benefits of systematic reviews in understanding the causes of interruption and designing future studies. This study provides implications encouraging future research directions regarding socio-psychological phenomena related to athletes’ motivation to continuously participate in physical activity. Exercise interruption intention reduced the re-participation intention; however, the path of YouTube content interaction in this regard was demonstrated when testing Hypothesis 5.

Social support had a significant indirect effect on re-participation intention to exercise via exercise performance satisfaction (H3). This result confirms the findings reported by Moreno-Murcia et al. [80], who studied 355 adult women and found that social support influenced life satisfaction. Another study on 216 elite athletes showed that social support significantly affected exercise performance satisfaction, also supporting the results of the present study [41]. Furthermore, Voltes-Dorta et al. [40] found that exercise performance satisfaction had a significant effect on exercise re-participation, further supporting the results of the current work. Hypothesis 4, that social support has a significant effect on re-participation intention by mediating the relationship between exercise performance satisfaction and exercise interruption intention, was rejected. In line with this finding, Park and Oh [81] studied 250 female participants in a yoga program and found that yoga satisfaction partially reduced exercise cessation, but they reported no significant effect. However, future research should strive to clarify the relationship between exercise performance satisfaction and the intention to give up exercise.

Hypothesis 5 stated that social support, exercise performance satisfaction, and exercise interruption intention significantly affect re-participation intention through the interaction of YouTube sports content. The results showed that YouTube sports content had a significant effect on re-participation intention at medium and high levels of social support and exercise interruption intention. In a previous study on 103 fitness YouTube users, Shen et al. [17] revealed that the participants acquired new movements and knowledge through YouTube videos. Furthermore, Sokolova et al. [16] found that YouTube viewing was significantly related to exercise participation motivation, supporting the results of this study. In particular, the researchers emphasized that the exercise effect appears when the exercise participants watch YouTube videos, excluding those who do not participate in sports. In other words, watching YouTube can serve as a surrogate experience that can partially replace exercise by providing the feeling of exercising. According to this study, there is a need for YouTube to be oriented so that exercise participants and those who have experienced exercise in the past can actively utilize it.

As suggested by Tiggemann and Zaccardo [82], this study is highly likely to indicate vicarious experiences because watching YouTube videos provides the feeling of actually exercising. In these cases, increased sedentary activity can lead to unhealthy behaviors [74]. People avoid vicarious experiences by watching YouTube. It is recommended that actual exercise participants and those who have experienced exercise use YouTube sports content viewing. Meanwhile, Wasan, Darmawan, and Kustandi [83] conducted a qualitative study on university students majoring in sports science and found that viewing YouTube content improved students’ learning motivation and creativity. Joo [84] studied 444 people and found that exercising while watching YouTube content positively affected participants’ health and exercise consciousness. YouTube sports content can enhance one’s participation in sports regardless of their location. This result aligns with the goal of sports for all in Korea.

In summary, this study designed a research model by intervening in YouTube sports content based on the sports socialization process model. The results indicate that physical activity participants experience exercise performance satisfaction and exercise re-participation intention through social support. In particular, the proposed research model found that exercise interruption intention reduces one’s re-participation intention YouTube sports content intervention. This part emphasizes the need to be dealt with in follow-up studies by specifically con-firming background variables (personality, age). Perhaps you could write, “Based on this study, the theoretical model of the sports socialization process is expected to intervene in YouTube sports content to extend current theories, provide directions for future research and intervention strategies, and increase the awareness of research methods. The understanding that the public can understand physical activity through YouTube sports content and that the continuous development of YouTube content and the expansion of physical activity participants provides vital implications from the socio-psychology point of view of sports.

## 5. Conclusions

This study extracted meaningful results through the intervention of YouTube sports content based on the sports socialization process model. However, this study has some limitations. For one, all study subjects participated in a specific competition. Thus, there may be limitations in generalizing the current findings to athletes who participate in physical activity in general. Future research needs to expand the group of research subjects to include athletes from different sports. Furthermore, the present study did not evaluate gender differences. Shen et al. [17] showed that men preferred strong exercise, while women reported that they preferred programs related to weight loss and body improvement. Thus, future studies should investigate the preferred physical activity program, motivation, and satisfaction according to gender. Joo [84] reports a preference for watching YouTube sports content for home training. In future studies, research to confirm the purpose of watching YouTube will be needed (e.g., skill improvement, lesson improvement, body maintenance, fun).

Most studies have used qualitative research methodologies to verify the sports socialization process [8,11,85]. The present study contributes to developing a sports socialization process model that was analyzed using a quantitative research methodology. A new model was discovered through the intervention of YouTube sports content in the process of socialization in sports. In this respect, the sports socialization process theory can be understood more comprehensively. Furthermore, the proposed model was verified using sports psychology variables, and it can contribute to the theoretical development of the convergence of sports socio-psychological. Future research should investigate the possibility of an omitted mediator between social support and re-participation intention [86].

Lastly, there is an urgent need to disseminate videos that can explain the importance of exercise to the public who do not exercise. Specifically, it is necessary to develop YouTube sports contents that can provide exercise participants with safe and effective training methods. Such content is expected to provide important information in the current era in which people prefer obtaining information through YouTube.

## Figures and Tables

**Figure 1 behavsci-13-00187-f001:**
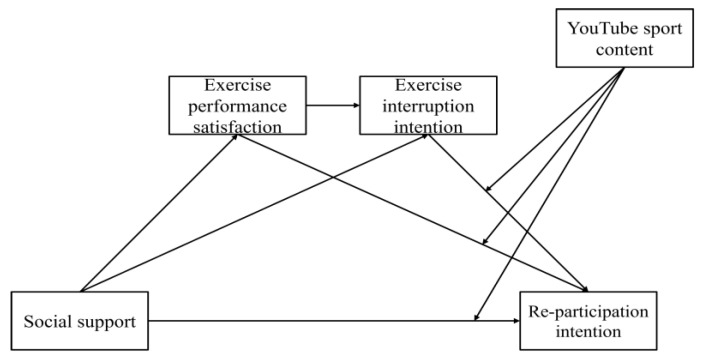
Finalized hypothesized model.

**Figure 2 behavsci-13-00187-f002:**
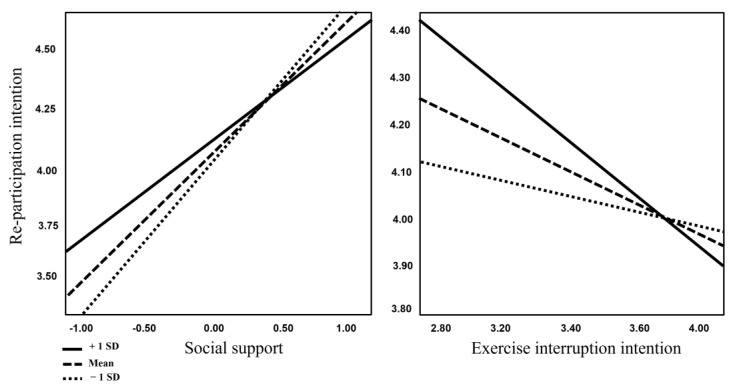
Interaction effects graph.

**Table 1 behavsci-13-00187-t001:** Descriptive statistics, Pearson correlations of scores, and Cronbach’s α.

Variable	1	2	3	4	5
1. Social support (1)	-	-	-		-
2. Exercise performance satisfaction (2)	0.497 **	-	-		-
3. Exercise interruption intention (3)	−0.508 **	−0.255 **	-		-
4. Re-participation intention (4)	0.733 **	0.467 **	−0.461 **	-	
5. YouTube sport content (5)	0.059	0.014	−0.182 **	0.101	-
Mean	4.23	3.46	1.80	4.09	3.75
Standard deviation	0.72	0.91	0.91	0.77	0.93
Skewness	−0.99	−0.29	1.39	−0.87	−0.85
Kurtosis	0.61	0.01	1.56	1.24	0.43
CR	0.93	0.87	0.96	0.88	0.94
AVE	0.60	0.57	0.68	0.56	0.54
Cronbach’s *α*	0.92	0.89	0.96	0.88	0.96

Note. CR: composite reliability; AVE: average variance extracted; ** *p* < 0.01.

**Table 2 behavsci-13-00187-t002:** Direct and moderated mediating effect index.

	B	S.E	t(p)	95% CI	
				LLCI	ULCI
Dependent Variable: Exercise performance satisfaction					
Constant	0.818	0.284	2.877 **	0.258	1.377
Social support	0.625	0.066	9.441 ***	0.495	0.756
Dependent Variable: Exercise interruption intention					
Constant	4.521	0.288	15.717 ***	3.954	5.087
Social support	−0.640	0.076	−8.404 ***	−0.789	−0.490
Exercise performance satisfaction	−0.004	0.061	−0.059	−0.123	0.115
Dependent Variable: Re-participation intention					
Constant	1.122	0.261	4.299 ***	0.608	1.636
Social support	0.650	0.056	11.589 ***	0.539	0.760
Exercise performance satisfaction	0.116	0.039	2.929 **	0.038	0.194
Exercise interruption intention	−0.101	0.039	−2.544 *	−0.179	−0.023
Total, Indirect effects	B	Boot S.E		BootLLCI	BootULCI
Total Indirect effect	0.139	0.070		0.014	0.294
Social support → Exercise interruption intention → Re-participation intention	0.073	0.033		0.014	0.144
Social support → Exercise performance satisfaction → Re-participation intention	0.065	0.032		0.009	0.136
Social support → Exercise performance satisfaction → Exercise interruption intention → Re-participation intention	0.001	0.005		−0.009	0.014

Note. *** *p* < 0.001, ** *p* < 0.01, * *p* < 0.05.

**Table 3 behavsci-13-00187-t003:** Interaction effect index.

	B	S.E	t(p)	95% CI	
				LLCI	ULCI
Dependent Variable: Re-participation intention					
YouTube sport content	0.048	0.034	1.407	−0.019	0.115
Social support × YouTube sport content	−0.118	0.060	−1.971 *	−0.235	−0.001
−0.926	0.694	0.071	9.804 ***	0.554	0.833
0.000	0.585	0.059	9.850 ***	0.468	0.702
0.926	0.476	0.090	5.275 ***	0.298	0.654
Exercise performance satisfaction × YouTube sport content	0.040	0.041	0.973	−0.041	0.120
Exercise interruption intention × YouTube sport content	−0.106	0.035	−3.042 **	−0.175	−0.037
−0.923	−0.063	0.043	−1.471	−0.147	0.021
0.000	−0.161	0.046	−3.534 ***	−0.251	−0.071
0.923	−0.259	0.067	−3.898 ***	−0.390	−0.128
Indirect effect: Social support → Exercise performance satisfaction → Re-participation intention					
−0.926	0.048	0.049		−0.051	0.146
0.000	0.071	0.033		0.012	0.140
0.926	0.094	0.040		0.026	0.182
Indirect effect: Social support → Exercise interruption intention → Re-participation intention					
−0.926	0.040	0.036		−0.035	0.107
0.000	0.103	0.036		0.037	0.181
0.926	0.166	0.061		0.066	0.303
Indirect effect: Social support → Exercise performance satisfaction → Exercise interruption intention → Re-participation intention					
−0.926	0.001	0.004		−0.007	0.010
0.000	0.001	0.008		−0.019	0.016
0.926	0.001	0.014		−0.034	0.023

Note. *** *p* < 0.001, ** *p* < 0.01, * *p* < 0.05.

## Data Availability

The data presented in this study are available on request from the corresponding author. The data are not publicly available due to privacy issues.

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
