# Peer review of "The Relationship between Exercise Re-Participation Intention Based on the Sports-Socialization Process: YouTube Sports Content Intervention"

_behavsci, 2023, doi:10.3390/bs13020187_

Round 1
Reviewer 1 Report
I believe this study is very well organized and it has a merit for this journal. Reliability and validity of the survey should be reported carefully. In addition, practical applications should be reported in the conclusion part. Best Regards.
Author Response
Dear.
Response to Reviewer 1 Comments.
Submit word file attachment.
Thankyou.

Reviewer 2 Report
The theme is very important and has a significant impact in today's world, as it relates physical exercise and technology. I really liked the methodology and the results were very interesting!!!
Congratulations to the authors... I just suggest a final review to remove small issues that still exist...
Author Response
Dear.
Response to Reviewer 2 Comments.
Submit word file attachment.
Thankyou.

Reviewer 3 Report
The authors speak of a socialisation model (36), but they don't discuss it. Is this a specific theoretical framework? The critique of the models and frameworks cited and the limitations given by the authors (49 and 50) need to be better approached. The analysis, at this stage, lacks detail and conceptual analysis. The authors assert their views without explaining and perspectives. The authors need to show the originality of their approach. Other quantitative studies exist but are not analysed or criticised in the text. (58-59) There is not just one study but several referenced by the authors. The authors must be careful not to be too prescriptive in their reasoning. They shouldn't try to confirm an already constructed position but rather question it to show the positive but also the negative effects of the internet media (and especially YouTube).
Citing authors and international studies is not enough, they must discuss the main lines of thought, concepts and currents of thought to show and illustrate the functions of socialisation, de-socialisation and re-socialisation of sports. The authors should further discuss the consequences and impacts of these studies in understanding the subject. This will allow the authors to better demonstrate the relevance of their study presented in this article (107-108).
Points 113 and 114 need to be clarified. The reader doesn't know whether the authors are referring to other studies or to their study. The authors need to be more specific in their heuristic approach. The links and correlations between the socialising effects of sport and the principle of social support are not clear in the text. The structuring of the article needs to be improved. The context of the study (YouTube and the internet) should be explained in one part, followed by a presentation of previous studies, showing their interests and limitations, and then focused to the questioning of the study in situ. The presentation is too confusing.
Globally, the article needs a more detailed and better constructed analysis. It is not easy to read and understand it. The methodology is not very explicit and the results are presented in an uneducational way. The authors don't really analyse the results of their study or discuss the impacts of YouTube as a social support to practice. The socio-psychological dimension is not really exposed or discussed. It is also necessary to add the questionnaire used in appendice with the main answers given by the interviewes.
Author Response
Dear.
Response to Reviewer 3 Comments.
Submit word file attachment.
Thankyou.

Round 2
Reviewer 3 Report
The authors has provided useful details in the introduction and also in the literature review on the socialisation process and the evolution of current research. Many references have been added in the text and in the bibliography. All this modifications make the study more relevant and underline the interest of the text. The figures are more explicit and easier to understand. The author has added relevant text in the discussion alway and also in the conclusion. Even if the text still has limitations on some elements of methodology and on the choice of the analysed sample (but the author himself acknowledges this), the text is better and more acceptable. I agree with publication in present form.